# Corneal Findings Associated to Belantamab-Mafodotin (Belamaf) Use in a Series of Patients Examined Longitudinally by Means of Advanced Corneal Imaging

**DOI:** 10.3390/jcm11102884

**Published:** 2022-05-19

**Authors:** Rita Mencucci, Michela Cennamo, Ludovica Alonzo, Carlotta Senni, Aldo Vagge, Lorenzo Ferro Desideri, Vincenzo Scorcia, Giuseppe Giannaccare

**Affiliations:** 1Department of Neurosciences, Psychology, Pharmacology and Child Health, Eye Clinic, University of Florence, 50134 Florence, Italy; rita.mencucci@unifi.it (R.M.); michelacennamo@libero.it (M.C.); ludovica.alonzo@gmail.com (L.A.); 2Department of Ophthalmology, University Vita-Salute, IRCCS Ospedale San Raffaele, 20132 Milan, Italy; senni.carlotta@hsr.it; 3IRCCS Ospedale Policlinico San Martino, University Eye Clinic of Genoa, 16132 Genova, Italy; aldo.vagge@gmail.com (A.V.); lorenzoferrodes@gmail.com (L.F.D.); 4Department of Neurosciences, Rehabilitation, Ophthalmology, Genetics, Maternal and Child Health (DiNOGMI), Università di Genova, 16126 Genova, Italy; 5Department of Ophthalmology, University Magna Græcia of Catanzaro, 88100 Catanzaro, Italy; vscorcia@unicz.it

**Keywords:** belantamab mafoditin, belamaf, multiple myeloma, cornea, side effects, vision

## Abstract

Belantamab mafodotin (belamaf) is a novel antibody–drug conjugate developed for the treatment of patients with relapsed or refractory multiple myeloma (RRMM). Although the drug has demonstrated a good efficacy, corneal adverse events have been reported. In this prospective study, consecutive patients with RRMM who received belamaf infusions were included. The standard ophthalmological visit was implemented with anterior segment (AS)-optical coherence tomography (OCT) and in vivo confocal microscopy (IVCM). Five patients (three males, two females; mean age 66 ± 6.0 years) with MMRR and unremarkable ocular findings at baseline who received belamaf infusion were included. After a median time of 28 days from the first infusion, four of them developed corneal alterations with transient vision reduction to a variable extent. In particular, corneal deposits of microcyst-like epithelial changes (MECs) were detected centrally in one patient and peripherally in three patients. AS-OCT scans showed a bilateral heterogeneous increase in signal intensity, together with hyper-reflective lesions confined within the epithelium in all cases, except for one case in which they also involved the stroma. Corneal maps showed a transient increase in epithelial thickness in the first phase that was followed by a diffuse decrease in the subsequent phase. IVCM scans showed MECs as hyper-reflective opacities located at the level of corneal epithelium, largely intracellular. Multimodal corneal imaging may implement the current clinical scale, helping us to detect corneal abnormalities in patients under belamaf therapy. This workup provides useful data for monitoring over time corneal findings and for optimizing systemic therapy.

## 1. Introduction

Ocular surface diseases are common among hematological patients, either due to the underlying diseases or the therapies employed for their treatment [1,2,3]. Multiple myeloma (MM) is a hematologic cancer characterized by uncontrolled proliferation and subsequent accumulation of malignant plasma cells in bone marrow. Despite the continuous attempt at improving management by investigating new classes of therapeutic agents, unfortunately, MM is still characterized by a poor prognosis. Indeed, a significant number of patients develop relapsed or refractory MM (RRMM) that is resistant to current standard-of-care options, pointing out the widespread problem of an unmet medical need [4].

Belantamab mafodotin (balamaf) is a first-in-class antibody–drug conjugate developed for the treatment of patients with RRMM. It consists of an anti-BMCA mAb conjugated to the microtubule inhibitor monomethyl auristatin F. Belamaf eliminates MM cells by multiple mechanisms of action, including apoptosis, antibody-dependent cell-mediated anti-myeloma responses, accompanied by the release of markers characteristic of immunogenic cell death [5]. Although the drug has demonstrated a deep and durable response in this type of patients, various corneal alterations have been reported so far, making the multidisciplinary management even more challenging.

Herein, we present a series of patients who developed corneal adverse events associated with belamaf use and describe features, clinical outcomes, and potential importance to oncologists as a harbinger of serious ocular sequelae. Furthermore, a multimodal imaging-based diagnostic workup used in this series to detect and monitor over time corneal alterations is presented.

## 2. Materials and Methods

This is a prospective observational study conducted at the Careggi University Hospital (Florence, Italy) that included consecutive patients who received belamaf infusions at the recommended dose (2.5 mg/kg) every 3 weeks for the treatment of RRMM between January 2021 and March 2021. All patients underwent an ophthalmological visit before starting treatment, including best corrected visual acuity (BCVA) testing, slit lamp evaluation of the cornea with photograph, intraocular pressure measurement and fundoscopy. The standard visit was implemented with a diagnostic workup focused on cornea that included anterior segment (AS)-optical coherence tomography (OCT) for the studying of corneal maps and in vivo confocal microscopy (IVCM). In particular, corneal epithelial thickness was measured in 5 sectors: one central 3 mm diameter and four inner sectors (inferior, superior, nasal, and temporal) within a ring (3–6 mm in diameter). Patients were followed-up for at least 12 months and underwent serial ophthalmological visits before each drug infusion.

## 3. Results

Overall, five patients (three males, two females; mean age 66 ± 6.0 years) with MMRR were screened for belamaf infusion during the study period. At baseline, all of them presented full vision (mean pre-belamaf BCVA of 20/20) and did not present any remarkable corneal findings; thus, they were treated with belamaf according to drug protocol. Following a mean time of 28 days from the first belamaf infusion, four patients developed corneal alterations with a variable degree of vision reduction (mean post-belamaf BCVA of 20/32). On slit lamp examination, corneal deposits with microcyst-like epithelial changes (MECs) were detected bilaterally in the center of the cornea in one patient and in the peripheral cornea in three patients (Figure 1). Of note, the patient with central MECs complained of blurred vision, while two of the remaining patients with peripheral MECs reported only dry-eye-like symptoms. Two patients switched to a modified therapy regimen, delaying by a week the treatment due to corneal adverse events. In two cases, the epithelial lesions, which were initially small and located in corneal periphery and mid-periphery, migrated towards central cornea, becoming progressively more numerous; afterwards, they slowly decreased in number, as demonstrated both clinically and instrumentally.

AS-OCT scans showed a bilateral heterogeneous increase in signal intensity, together with hyper-reflective lesions corresponding to visible corneal alterations at slit lamp. These lesions were confined within the epithelium, but in one case, they migrated deeper, involving the stroma (Figure 2).

Corneal epithelial maps showed a transient CET increase soon after the first infusion, followed by a progressive decrease during the following infusions (“plateau phase”) (Figure 3 and Figure 4).

Using IVCM, MECs appeared as hyper-reflective opacities, sometimes arranged in clusters, resembling a “pseudo-rosette” pattern, located at the level of corneal epithelium, largely intracellular. In particular, in one patient, multiple hyper-reflective deposits were detected inside corneal epithelium at the level of alar and basal corneal cells rather than superficial cells; in another patient, completely asymptomatic, isolated peripheral sub-epithelial opacities were detected that subsequently were also found at the level of anterior stroma by means of IVCM (Figure 5).

## 4. Discussion

Since the introduction of belamaf therapy for RRMM, a number of corneal side effects have been attributed to its use. These mostly include superficial punctate keratopathy and/or MECs detectable at slit lamp examination. Of note, ocular adverse events such as blurred vision, dry-eye-like symptoms and keratopathy have been reported in 69% to 74% of patients in the DREAMM-1 and DREAMM-2 trials, requiring dose reduction or even interruption in 27% to 46% of cases [6,7]. Remarkably, other off-label drugs previously adopted for the treatment of MM were shown to induce corneal epithelial alterations, probably resulting from drug accumulation in the epithelium and subsequent cellular apoptosis. Among these, depatuximab mafoditin, which acts as an epidermal growth factor receptor inhibitor, caused a reversible corneal epitheliopathy in treated patients, characterized by multiple and diffuse hyper-reflective spots, progressive sub-basal nerve plexus fragmentation and by the appearance of cystic structures at the level of corneal epithelium [8,9,10].

Belamaf-related corneal alterations usually originate in the peripheral region of the cornea and then gradually migrate towards the center (“centripetal pattern”) [4]. The localization of corneal opacities in the central cornea along with the related irregular astigmatism determine changes in vision, including subjective blurred vision. However, a linear relationship between the severity of keratopathy and the reduction in vision has not been observed. More recently, corneal staining patterns suggestive of limbal stem cell deficiency have been described in a cohort of patients receiving prolonged therapy [11].

Corneal toxicity is more likely to occur when the drug is used at higher doses, and the proposed mechanism is thought to derive from the off-target effect of its cytotoxic component (i.e., the microtubule-disrupting monomethyl auristatin-F [MMAF]) [6].

Efforts are being made to understand the etiology behind this clinical entity in order to identify potential mitigation strategies. For this task, multimodal imaging is of crucial importance, as it helps to better characterize the nature of such findings. Of note, the involved areas of corneal epithelium were found to contain hyper-reflective material, rather than microcysts on IVCM. The solid nature of these epithelial lesions is also corroborated by AS-OCT, where they appear once again hyper-reflective. Therefore, this material could potentially represent the accumulation of apoptotic end products in the intercellular spaces of corneal epithelium. Similarly, hyper-reflective lesions on AS-OCT may be the consequence of the accumulation of either pre-apoptotic or degenerated cells caused by the internalization of belantamab [11]. Specifically, it has been hypothesized that belamaf could reach the cornea through limbal vessels or tears and could then undergo cellular uptake into the corneal basal epithelial layer. Once internalized into epithelial cells, it may induce their apoptosis via microtubulin inhibition during their travel from the basal layer towards the surface and the center of cornea [12]. Nevertheless, belamaf deposits were identified also in the sub-Bowman layer, suggesting that other mechanisms of toxicity might also be involved [13].

In order to provide a common standardized and repeatable tool for clinical evaluation, an expert board has recently proposed a keratopathy and visual acuity scale (KVA scale) based on the worst finding of either keratopathy subjectively graded on slit lamp examination (the deeper is the corneal involvement the worst is the grade of the keratopathy) or visual acuity testing. Recommendations for the management of corneal events have been formulated according to the KVA scale as follows: continue treatment at the current dose for grade 1/mild events; delay treatment for grade 2/moderate events until the event improves to a grade 1/mild event or resolves, or for grade 3/4 (severe) events, until these improve to a grade 1/mild event [6].

Our report aimed at describing corneal findings of patients receiving belamaf infusions for RRMM, especially focusing on clinical course monitored by different corneal imaging techniques. Thanks to advances in multimodal corneal imaging including IVCM and AS-OCT, ophthalmologists may benefit from these techniques by easily detecting corneal abnormalities, even at a subclinical stage, and subsequently defining and monitoring over time the entity of corneal involvement [11,12,13,14].

Using IVCM, MECs appear as hyper-reflective opacities, sometimes arranged in a “pseudo-rosette” pattern, located within the basal epithelium and the sub-basal nerve plexus; the stroma shows only non-specific signs of inflammation with sporadic activated keratocytes, while the endothelium is typically unaffected; specific lesions at the sub-basal nerve plexus layer were also recently detected and characterized [13]. However, although IVCM is very sensitive to identify potential preclinical findings, it does not provide quantitative measurements of biomarkers useful for monitoring over time corneal changes, especially when few modifications must be detected. Using AS-OCT, corneal alterations can be detected as hyper-reflective lesions, usually located within the corneal epithelium. In one case of our series, it allowed for the identification of lesions located deeper in the stroma (Figure 2). To the best of our knowledge, this AS-OCT finding is reported here for the first time. Furthermore, corneal maps allowed us to measure the thickness of the corneal epithelium that was found to be increased in the initial phase of the keratopathy and then tended to decrease to baseline values (Figure 3) [11].

However, it should be pointed out that, since half of the patients may present objective signs of keratopathy without reporting any symptoms, ophthalmic examination should be performed at baseline and before each dose of the drug in order to detect early corneal changes.

The management of belamaf-induced toxicity requires a multidisciplinary approach involving a hematologist and an ophthalmologist, and the clinical decision should be reached together taking into account the individual benefit/risk ratio according to both ocular and systemic conditions. Overall, single-agent belamaf (2.5 mg/kg) has shown a manageable safety profile. Ocular alterations usually recover spontaneously once treatment is discontinued or may be successfully reversed with dose tapering or by delaying the time interval of suspension between two cycles. This evidence is based on the experience of a DREAMM-2 study that reported the need for drug discontinuation due to keratopathy in only 1 out 95 patients who received the 2.5 mg/kg dose, indicating that patients were able to remain on treatment while these events were monitored [5]. In this regard, the recent flow chart of multidisciplinary approach created by an expert board for managing corneal events with belamaf will assist physician to better cope with this condition [6]. The implementation of the KVA scale with a multimodal corneal imaging may further help to detect corneal abnormalities, even at a subclinical stage, providing useful data for monitoring over time corneal findings and for optimizing belamaf therapy. Although both AS-OCT and IVCM represent two valuable tools for this task, the former examination and, in particular, corneal topography with specific focus on the epithelial map could be incorporated in the current clinical scale (KVA) in order to quantify corneal involvement and provide a reliable biomarker. Furthermore, unlike IVCM, this technique is completely noninvasive and is available in almost the totality of ophthalmic departments.

Future prospective trials are needed to provide more comprehensive information about belamaf-associated keratopathy as well as for identifying the most appropriate diagnostic workup able to optimize therapy in RRMM patients.

## Figures and Tables

**Figure 1 jcm-11-02884-f001:**
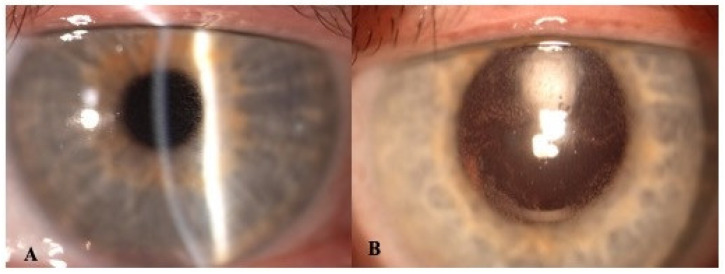
Slit-lamp photograph of the cornea in a patient with belamaf keratopathy. (**A**): Diffuse microcystic-like epithelial changes (MECs) in the central cornea. (**B**): Better visualization of MECs aided by retroillumination and pupil dilation.

**Figure 2 jcm-11-02884-f002:**
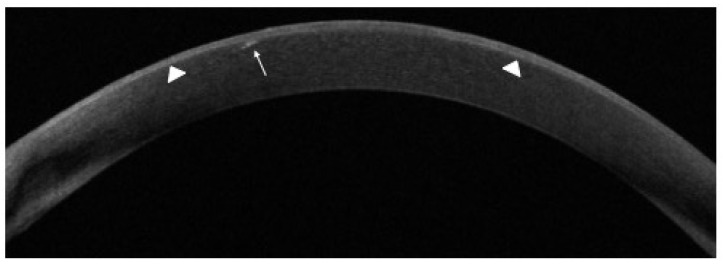
Anterior segment optical coherence tomography (AS-OCT) in a patient with belamaf keratopathy showing heterogeneous, diffuse corneal epithelium hyper-reflectivity corresponding to MECs (arrowheads), with a small hyper-reflective area extending to the anterior stroma associated with a focal interruption of the Bowman layer (arrow).

**Figure 3 jcm-11-02884-f003:**
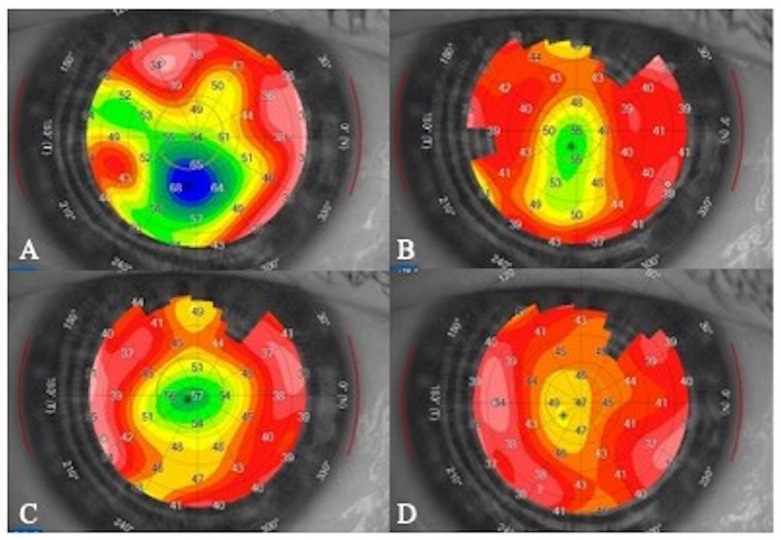
Corneal epithelial thickness (CET) mapping during monthly infusions of belamaf in a patient with belamaf keratopathy. (**A**): Before the second infusion showing a localized transitory CET increase; (**B**–**D**): Progressive reduction in CET over time, respectively, before the third, fourth and fifth infusion (“plateau phase”).

**Figure 4 jcm-11-02884-f004:**
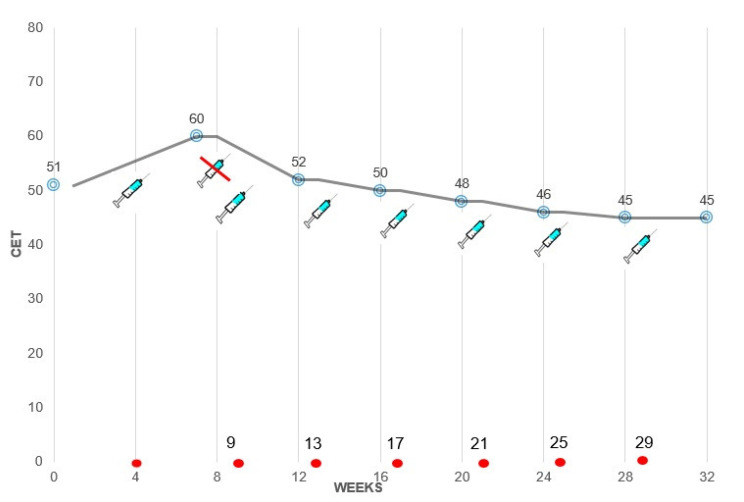
Graph showing the changes of average values of corneal epithelial thickness (CET) measured in 5 sectors according to the infusion of belamaf in a representative patient. After the first dose, an increase in CET with the appearance of central MECs and vision reduction were found. The second dose was therefore delayed by one week. After the third and the fourth dose, a diffuse decrease in CET was assessed, followed by a “plateau phase” after the fifth dose. CET values are expressed in micron.

**Figure 5 jcm-11-02884-f005:**
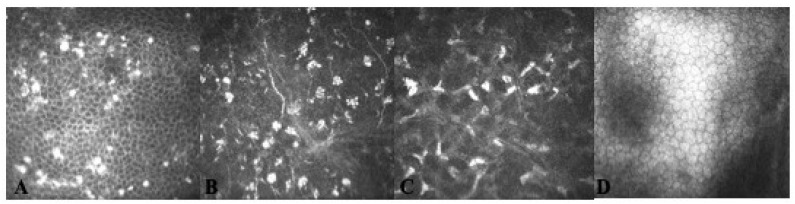
In vivo confocal microscopy (IVCM) of the cornea in a patient with belamaf keratopathy. (**A**): MECs appeared as hyper-reflective (at least predominantly) intracellular opacities present at the level of basal epithelium. (**B**): Hyperreflective lesions arranged in clusters, resembling “pseudo-rosette” pattern at the level of sub-basal nerve plexus. (**C**): Activated keratocyte network detected in the anterior stroma. (**D**): Normal endothelial cells morphology.

## Data Availability

The data presented in this study are available on request from the corresponding author. The data are not publicly available due to privacy issues.

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
