# Peer review of "Corneal Findings Associated to Belantamab-Mafodotin (Belamaf) Use in a Series of Patients Examined Longitudinally by Means of Advanced Corneal Imaging"

_jcm, 2022, doi:10.3390/jcm11102884_

Round 1

Reviewer 1 Report

I read with great interest this paper and I really think it is nice, well written and important. I truly believe that report these complications are necessary for our community since many of us follow up these patients in several RCT

Some recommendations:

Since MM has been treated with other off label drugs previously to belantamab and this could also produce corneal cyst or keratopathy, I would mention it (depatuxizumab) in the introduction or discussion (DOI: 10.5935/0004-2749.20220039/doi:10.1097/ICO.0000000000002595/doi:10.3389/fonc.2020.593461.)

Minor changes:

Type error: (BVCA)

Author Response

I read with great interest this paper and I really think it is nice, well written and important. I truly believe that report these complications are necessary for our community since many of us follow up these patients in several RCT.

Thanks for your positive comments.

Some recommendations:

Since MM has been treated with other off label drugs previously to belantamab and this could also produce corneal cyst or keratopathy, I would mention it (depatuxizumab) in the introduction or discussion (DOI: 10.5935/0004-2749.20220039/doi:10.1097/ICO.0000000000002595/doi:10.3389/fonc.2020.593461.)

Thanks for the suggestion. We added the following sentences in the discussion section: Remarkably, other off-label drugs previously adopted for the treatment of MM were shown to induce corneal epithelial alterations, probably resulting from drug accumulation in epithelium and subsequent cellular apoptosis. Among these, depatuximab mafoditin, which acts as an epidermal growth factor receptor inhibitor, caused a reversible corneal epitheliopathy in treated patients, characterized by multiple and diffuse hyper-reflective spots, progressive sub-basal nerve plexus fragmentation and by the appearance of cystic structures at the level of corneal epithelium.8-10”

Minor changes:

Type error: (BVCA)

Ok corrected.

Reviewer 2 Report

In their manuscript entitled „Corneal Findings Associated to Belantamab-Mafodotin (Belamaf) Use in a Series of Patients Examined Longitudinally by Means of Advanced Corneal Imaging “, authors Mencucci et al. report clinical data from 5 patients who developed corneal epithelial (and anterior stromal) alterations following administration of a novel drug for the treatment of multiple myeloma. This is a novel and interesting topic because the Belantamab-Mafodotin is very much welcomed by oncologists as an effective treatment in this malignant condition. However, ocular side-effects are extremely common and can be quite debilitating for affected patients; hence, it is of great interest to characterize this new condition. Also, findings from this patient population may enhance our understanding of corneal surface regeneration.

I applaude the authors for using the term “microcyst-like epithelial changes”; they may want to emphasise that their findings corroborate the notion that the changes that appear like cysts on slit-lamp examination are not actually cystic, but rather hyper-reflective (hence solid) on ocular imaging. The authors may want to discuss what this may mean for our understanding of the potential pathogenetic mechanisms behind this clinical entity, i.e. epithelial cell necrosis.

A major weakness of this manuscript is that the findings are not entirely novel, although this case-series is very well-described. However, it would be beneficial if the manuscript would acutally meet its claim of a longitudinal examination of these patients by demonstrating the dynamics of corneal changes over time and correlating this to dosage of the drug and clinical effects on the underlying condition.

Lines 175-179: “Ocular alterations rarely recover spontaneously” – this statement is misleading. If the drug is discontinued, ocular alterations do indeed resolve spontaneously, unless ulceration or frank perforation have occurred. “…reversed with dose escalation…” – increasing the dose, however, is unlikely to improve the corneal situation. “…treatment interruption (…) can lead to poorer clinical outcomes…” – please provide a reference! To my knowledge, Patients that have marked ocular side-effects also tend to respond well to the drug, hence interrupting treatment until the corneal surface improves does not compromise the desired effect.

Author Response

In their manuscript entitled “Corneal Findings Associated to Belantamab-Mafodotin (Belamaf) Use in a Series of Patients Examined Longitudinally by Means of Advanced Corneal Imaging “, authors Mencucci et al. report clinical data from 5 patients who developed corneal epithelial (and anterior stromal) alterations following administration of a novel drug for the treatment of multiple myeloma. This is a novel and interesting topic because the Belantamab-Mafodotin is very much welcomed by oncologists as an effective treatment in this malignant condition. However, ocular side-effects are extremely common and can be quite debilitating for affected patients; hence, it is of great interest to characterize this new condition. Also, findings from this patient population may enhance our understanding of corneal surface regeneration.

Thanks for your positive comments. 

I applaude the authors for using the term “microcyst-like epithelial changes”; they may want to emphasise that their findings corroborate the notion that the changes that appear like cysts on slit-lamp examination are not actually cystic, but rather hyper-reflective (hence solid) on ocular imaging. The authors may want to discuss what this may mean for our understanding of the potential pathogenetic mechanisms behind this clinical entity, i.e. epithelial cell necrosis.

Thanks for the suggestion. We agree with you about the importance of understanding the pathogenetic mechanisms behind this clinical entity. In this regard, we added a new paragraph in the discussion section as follows: “Efforts are being made to understand the etiology behind this clinical entity in order to identify potential mitigation strategies. For this task, multimodal imaging is of crucial importance as it helps to better characterize the nature of such findings. Of note, involved areas of corneal epithelium were found to contain hyper-reflective material, rather than microcysts on IVCM. The solid nature of these epithelial lesions is also corroborated by AS-OCT where they appear once again hyper-reflective. Therefore, this material could potentially represent the accumulation of apoptotic end products in the intercellular spaces of corneal epithelium. Similarly, hyper-reflective lesions on AS-OCT may be the consequence of the accumulation of either pre-apoptotic or degenerated cells caused by internalization of belantamab.11”

A major weakness of this manuscript is that the findings are not entirely novel, although this case-series is very well-described. However, it would be beneficial if the manuscript would actually meet its claim of a longitudinal examination of these patients by demonstrating the dynamics of corneal changes over time and correlating this to dosage of the drug and clinical effects on the underlying condition.

Thanks for the intriguing suggestion. We believe that currently our data are too limited to conduct a reliable correlation analysis between the dosage of the drug and corneal side effects. However, we created a new figure (Figure 4) containing a graph that shows the changes of average corneal epithelial thickness measured in 5 sectors according to the infusion of belamaf in a representative patient. After the first dose an increase of CET with the appearance of central MECs and vision reduction were found. The second dose was therefore delayed by one week. After the third and the fourth dose, a diffuse decrease of CET was assessed followed by a “plateau phase”after the fifth dose.

Lines 175-179: “Ocular alterations rarely recover spontaneously” – this statement is misleading. If the drug is discontinued, ocular alterations do indeed resolve spontaneously, unless ulceration or frank perforation have occurred. “…reversed with dose escalation…” – increasing the dose, however, is unlikely to improve the corneal situation. “…treatment interruption (…) can lead to poorer clinical outcomes…” – please provide a reference! To my knowledge, Patients that have marked ocular side-effects also tend to respond well to the drug, hence interrupting treatment until the corneal surface improves does not compromise the desired effect.

Thanks for this suggestion that allow us to be clearer about this point. We added in the discussion section the following sentences: Overall, single-agent belamaf (2.5mg/kg) has shown a manageable safety profile. Ocular alterations usually recover spontaneously once treatment is discontinued or may be successfully reversed with dose tapering or by delaying the time interval of suspension between two cycles. This evidence is based on the experience of DREAMM-2 study that reported the need for drug discontinuation due to keratopathy in only 1 out 95 patients who received the 2.5-mg/kg dose, indicating that patients were able to remain on treatment while these events were monitored.5”

Round 2

Reviewer 1 Report

Authors have update the manuscript. According to me it is interesting for our community. Congratulations